# The Benchmark Lottery

**Mostafa Dehghani**[*], **Yi Tay**[*], **Alexey A. Gritsenko**[*], **Zhe Zhao, Neil Houlsby,**
**Fernando Diaz, Donald Metzler**[†]**, Oriol Vinyals**[†]
Google Research & DeepMind
{dehghani, yitay, agritsenko}@google.com

## Abstract

The world of empirical machine learning (ML) strongly relies on benchmarks in order to determine the relative effectiveness of different algorithms and methods. This paper proposes the notion of *a benchmark lottery* that describes the overall fragility of the ML benchmarking process. The benchmark lottery postulates that many factors, other than fundamental algorithmic superiority, may lead to a method being perceived as superior. On multiple benchmark setups that are prevalent in the ML community, we show that the relative performance of algorithms may be altered significantly simply by choosing different benchmark tasks, highlighting the fragility of the current paradigms and potential fallacious interpretation derived from benchmarking ML methods. Given that every benchmark makes a statement about what it perceives to be important, we argue that this might lead to biased progress in the community. We discuss the implications of the observed phenomena and provide recommendations on mitigating them using multiple machine learning domains and communities as use cases, including natural language processing, computer vision, information retrieval, recommender systems, and reinforcement learning.

## 1 Introduction

Quantitative evaluation is a cornerstone of machine learning research. As a result, benchmarks, including those based on data sets and simulations, have become fundamental to tracking the progress of machine learning research. Benchmarks have a long history in artificial intelligence research generally. There have been several attempts at designing milestones to capture progress toward artificial intelligence (e.g., human level game performance, the Turing test [Turing, 1950]). Specific system properties are measured through specialized benchmarks (e.g. for vision, natural language processing, robotics). All of these benchmarks, by design, encode values about what is salient and important, both across domains (e.g. natural language processing benchmarks versus robotics benchmarks) and within them (e.g. which languages are considered in an NLP benchmark, which environments are considered in a robotics benchmark).

As benchmarks become widely accepted, researchers adopt them, often without questioning their assumptions, and algorithmic development becomes slowly tied to these success metrics. Indeed, over time, the research community makes collective decisions about what shared tasks–and values–are important (through peer review norms and resource investment) and which are not.

Because of this, it is important for the research community to understand the individual, community, social, and political pressures that influence why some benchmarks become canonical and others do not. This paper shares some opinions on this topic along with case studies calling for discussion and reconsiderations on several issues with benchmarking in machine learning and argues that a meta-level understanding of benchmarks is a prerequisite for understanding how the progress in machine learning is made. This paper presents analyses on how benchmarks may affect the direction and pace of progress in machine learning and puts forward the notion of a benchmark lottery. We argue that many factors other than the algorithmic superiority of a method may influence the emergence of algorithms that are perceived as better. Moreover, we claim that for a method to emerge successful, it has to first win the

---

[*]Equal contribution, [†]equal advising.

Submitted to the 35th Conference on Neural Information Processing Systems (NeurIPS 2021) Track on Datasets and Benchmarks. Do not distribute.

*benchmark lottery*. Out of the many potential trials in this lottery, a method has to be first well-aligned with the suite of benchmarks that the community has accepted as canonical. We refer to the alignment between the tasks brought forth by the community and successful algorithms as the *task selection* bias. We empirically show that the task selection process has a great influence over the relative performance of different methods. Moreover, we argue that benchmarks are *stateful*, meaning that the method has to also participate in the lottery at the right moment, and to align well with existing techniques, tricks, and state-of-the-art. Related to this, we also briefly discuss how benchmark reuse may affect the statistical validity of the results of new methods.

As a whole, as we researchers continue to participate in the benchmark lottery, there are long-term implications, which we believe are important to be explicitly aware of. As such, the main goals of this paper are to (i) raise awareness of these phenomena and potential issues they create; and to, (ii) provide some recommendations for mitigating these issues. We argue that community forces and task selection biases, if left unchecked, may lead to unwarranted overemphasis of certain types of models and to unfairly hinder the growth of other classes of models - which may be important for making fast and reliable progress in machine learning.

The notion of what makes a benchmark canonical, in the sense that is widely accepted by the community, is also diverse depending on the field of study. On one hand, fields like natural language processing (NLP) or computer vision (CV) have well-established benchmarks for certain problems. On the other hand, fields such as recommender systems or reinforcement learning tend to allow researchers more freedom in choosing their own tasks and evaluation criteria for comparing methods. We show how this may act as *rigging the lottery*, where researchers can "make their own luck" by fitting benchmarks and experimental setups to models instead.

Overall, this paper explores these aspects of model evaluation in machine learning research. We frame this from a new perspective of the *benchmark lottery*. While there has been recent work that peers deeply into the benchmark tasks themselves [Bowman and Dahl, 2021], this work takes meta- and macro-perspectives to encompass factors that go beyond designing reliable standalone tasks.

The remainder of the paper is organized as follow: Section 2 discusses how benchmarks can influence long-term research directions in a given (sub-)field. Section 3 introduces the *task selection bias* and using established benchmarks as examples shows how relative performance of algorithms is affected by the task selection process. Section 4 takes another view of the task selection bias and proposes *community bias* as a higher-level process that influences task selection. We show that forces from the broader research community directly impact the task selection process and as a result, play a substantial role in creating the lottery. Section 5 posits that benchmarks are stateful entities and that participation in a benchmark differs vastly depending upon its state. We also argue continual re-use of the same benchmark may be problematic. Section 6 discusses *rigging the lottery*, the issue that some communities (e.g. recommender systems and reinforcement learning) face, where the lack of well-established community-driven sets of benchmarks or clear guidelines may inadvertently enable researchers to fit benchmarks to model. We highlight the potential drawbacks of such an approach. Finally, in Section 7 we provide recommendations for finding a way out of the lottery by building better benchmarks and rendering more accurate judgments when comparing models.

Overall, unified benchmarks have led to incredible progress and breakthroughs in machine learning and artificial intelligence research [Kingma and Welling, 2013, Mikolov et al., 2013, Sutskever et al., 2014, Bahdanau et al., 2014, Goodfellow et al., 2014, Hinton et al., 2015, Silver et al., 2016, He et al., 2016a, Vaswani et al., 2017, Devlin et al., 2018, Brown et al., 2020, Dosovitskiy et al., 2020]. There is certainly a lot of benefits of having the community come together to solve shared tasks and benchmarks. Given that the role of benchmarks is indispensable and highly important for measuring progress, this work seeks to examine, introspect and find ways to improve.

## 2  Background

Measuring progress is one of the most difficult aspects of empirical computer science and machine learning. Such questions as "What are the best setup and task to use for evaluation?" [Ponce et al., 2006, Machado et al., 2018, Lin, 2019, Bowman and Dahl, 2021, Recht et al., 2019, Lin et al., 2021, Gulcehre et al., 2020, Perazzi et al., 2016, Vania et al., 2020, Musgrave et al., 2020], "Which data or benchmark are most applicable?" [Metzler and Kurland, 2012, Beyer et al., 2020, Northcutt et al., 2021, Gulcehre et al., 2020, Dacrema et al., 2019], "Which metrics are suitable?" [Machado et al., 2018, Bouthillier et al., 2021, Balduzzi et al., 2018, Bouthillier et al., 2019, Musgrave et al., 2020], or "What are the best practices for fair benchmarking?" [Torralba and Efros, 2011, Armstrong et al., 2009, Machado et al., 2018, Sculley et al., 2018, Lin, 2019, Bowman and Dahl, 2021, Bouthillier et al., 2021, Recht et al., 2019, Lin et al., 2021, Balduzzi et al., 2018, Lipton and Steinhardt, 2018, Bouthillier et al., 2019, Vania et al., 2020, Mishra and Arunkumar, 2021, Marie et al., 2021, Dodge et al., 2019]

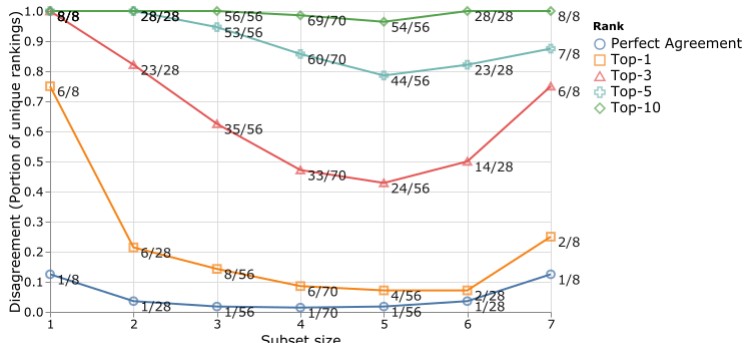

Figure 1: Disagreement of model rankings on the SuperGLUE benchmark as a function of the number of selected benchmark tasks. The $x$-axis represents the number of tasks in each sub-selection of tasks and each line corresponds to a different value of $k$ for the Top-$k$ in the rankings. Points are labels as $A/B$, where $A$ is the number of unique model rankings and $B$ is the total number of possible task combinations for this subset size. If $A = 1$, then all rankings are equivalent and consistent across all task selections; higher values of $A$ correspond to higher degrees of disagreement between models rankings.

are of utmost importance to correct empirical evaluation of new ideas and algorithms, and have been extensively studied. Nevertheless, the jury is still out on most of these questions.

We argue that some models and algorithms are not inherently superior to their alternatives, but are instead perceived as such by the research community due to various factors that we discuss in this paper. One of these factors is the software and hardware support for an idea, as captured in the concept of hardware lottery by Hooker [2020]. Here however we focus mainly on *benchmarking*-related factors, and discuss the role they play in the selection of a model as "fashionable" in the research world, and how this is often conflated with the model being better. When a class of models or algorithms gets recognition in the community, there will be more follow up research, adaption to more setups, more tuning and discovery of better configurations, which lead to better results. This is a valid way of propelling the field further. However, a question that we should also ask is how much progress could have been made by investing the same amount of time, effort, computational resources and talent in a different class of models. In other words, assuming model development as a complex high-dimensional optimization process, in which researchers are exploring a fitness surface, the initial point, as well as the fitness function, are the key factors for ending up with better optima, and both these factors are highly affected by the benchmarks used for evaluation.

## 3 Task selection bias

As we show in this section, relative model performance is highly sensitive to the choice of tasks and datasets it is measured on. As a result, the selection of well-established benchmarks plays a more important role than is perhaps acknowledged, and constitutes a form of partiality and bias - the *task selection bias*.

### 3.1 Case Studies

In this section, we study different popular benchmarks and use the data from the leaderboards of these benchmarks to run analyses that highlight the effect of task selection bias.

### 3.1.1 SuperGLUE

In order to study the effect of aggregated scores and how findings change by emphasizing and de-emphasizing certain tasks, we explore the SuperGLUE dataset [Wang et al., 2019]. To demonstrate the task selection bias on this benchmark, we re-compute the aggregated scores using different combinations of eight SuperGLUE tasks. We consider over $55$ different top performing models that are studied in [Narang et al., 2021], including transformer-based models with various activation functions, normalization and parameter initialization schemes, and also architectural extensions (e.g., Evolved Transformers [So et al., 2019], Synthesizers [Tay et al., 2020a], Universal Transformer [Dehghani et al., 2019], and Switch Transformers [Fedus et al., 2021]) as well as convolution-based models (e.g. lightweight and dynamic convolutions). We consider the fine-grained scores of these models on the $8$ individual tasks of SuperGLUE and their different combinations. For each combination of tasks, we take a mean-aggregate model performance for all models on the selected tasks and produce a ranking of all $55$ models. To make this ranking more meaningful, we only consider its Top-$k$ entries, where $k \in \{1, 3, 5, 10\}$.

**Ranking inconsistency.** Figure 1 gives a concise overview of the number of unique Top-$k$ rankings produced obtained from fixed-size subsets of tasks. For example among the $70$ different possibilities of selecting $4$ out of $8$ tasks, there are $6$ distinct model ranking orders produced for Top-1 (i.e. there are $6$ different possible top models). Moreover, when considering Top-3 or even Top-5, almost $60$ out of $70$ rankings do not agree with each other. Overall, the rankings become highly diverse as the subset of tasks selected from the benchmark is varied. This forms the core of the empirical evidence

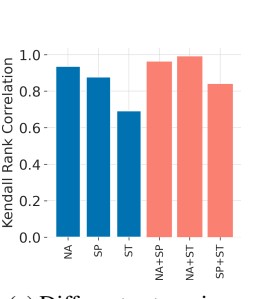 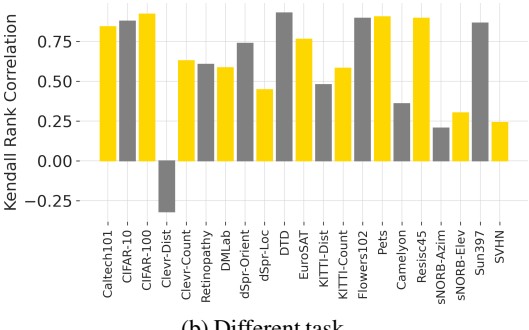

(a) Different categories.                    (b) Different task.

Figure 2: Rank correlation between the full VTAB score and the score for subsets of the benchmark.

of the task selection bias. More analyses on ranking of models on all possible combinations of tasks, rank correlation between SuperGLUE score and individual tasks, effect of relative raking of models in Appendix A.1,A.2, and A.3.

### 3.1.2 Visual Task Adaptation Benchmark (VTAB)

A similar situation can be observed for the Visual Task Adaptation Benchmark (VTAB; [Zhai et al., 2019]) benchmark. VTAB is used for evaluating the quality of representations learned by different models in terms of their ability to adapt to diverse, unseen tasks with few examples. VTAB defines a total of 19 tasks, grouped into three categories: *Natural*, *Specialized*, and *Structured*. We have evaluated 32 different models against all the 19 VTAB tasks. The difference between models is on their architectures (e.g. WAE-GAN [Tolstikhin et al., 2017] vs. VIVI[Tschannen et al., 2020]), their sizes (e.g. ResNet-50 vs. ResNet-101 [Kolesnikov et al., 2019]), or the dataset they were pre-trained on (e.g. ResNet-50 pretrained on ImageNet-21k vs. ResNet-50 pretrained on JFT [Kolesnikov et al., 2019]). Models we considered in our study are those that are introduced as "representation learning algorithms" in [Zhai et al., 2019]. More details on the tasks, categories, and models can be found in Appendix A.4.

First, we study the agreement of the aggregated score across all 19 tasks with the aggregated scores obtained from different combinations of the three task categories: natural (NA), specialized (SP), and structured (ST). Figure 2a shows the Kendall rank correlation, when ranking different models based on the full VTAB score and based on the category (combination) score. It can be seen that rankings of models based on different combinations of categories are not always perfectly correlated. For instance, the structured (ST) subcategory has a correlation of $\approx 0.7$ with the full VTAB score, thus highlighting rather different aspects of the competing models. A more striking point is the full disagreement of different subcategories on the winning model, i.e. top-1 that is shown in Appendix B, where we future present the results that show disagreement in the top-1, 2, and 3 rank positions based on different combinations of sub-categories and tasks. This shows that crowning a model as the winner based on a single score can be suboptimal, and demonstrates how the random nature of task selection can become a lottery that algorithms need to win.

Figure 2b also presents the correlations between the rankings based on the individual tasks and the aggregated VTAB score. Unsurprisingly, an even stronger disagreement between rankings is observed (mean Kendall correlation of $\approx 0.60$), including tasks with negative correlation. For more analyses and additional case studies (Long Range Arena and RL-Unplugged) check Appendix A.

### 3.2 Score and rank aggregation

So far, we highlighted the issue with reporting a single aggregated score that is supposed to reflect the performance on multiple tasks, by showcasing the disagreement between different subsets of tasks. One of the main difficulties for aggregating scores of multiple tasks is the lack of a clear mechanism for incorporating the difficulty of tasks into account. This is made more complex by the fact that there are multiple facets to what makes a task difficult. For instance, the size of the training data for different tasks, the number of prediction classes (and consequently the score for a random baseline for the task), distribution shift between the pretraining dataset and the downstream tasks, different performance ranges across tasks, or overrepresenting particular aspects by multiple tasks that introduces biases into averages [Balduzzi et al., 2018]. As a concrete example, in the case of VTAB some tasks use the same input data thus upweighting those domains, e.g. CLEVR-Count and CLEVR-Dist use the same data for different tasks, and for this particular example, given the negative correlation between CLEVR-Dist and the mean score, this upweighting effect makes the aggregated score even noisier.

To address some of these issues, there are alternative ways for ranking models instead of using the mean score across all tasks as the model performance on the benchmark. For instance, One can grouping tasks based on their domain) and use macro-averaging to account for the effect upweighting some domains [Zhai et al., 2019]. Given that using simple averaging for aggregation across multiple tasks, the maximum score is bounded, this may limit the range of performances, implicitly upweighting tasks with more headroom. To address this issue, one can use geometric mean instead of arithmetic mean.

There are also solutions for rank aggregation that ignore absolute score differences in favor of relative ordering [Dwork et al., 2001, Tabrizi et al., 2015]: For instance, the "average rank" that is obtained by ranking the methods for each task based on their score and then computing the average ranks across tasks. Another alternatives are, for instance, robust average rank, where, before averaging ranks across tasks, the accuracy is binned into buckets of size 1% and all methods in the same bucket get the same rank or elimination ranking (which is equivalent to an exhaustive ballot voting system) [Hao and Ryan, 2016].

### 3.3 Human evaluation bias

Related to the task selection bias we discussed in this section, *human evaluation bias* within a task can also play a role in model selection in some tasks like natural language generation. Lack of consistency in how human evaluation, e.g. due to different levels of expertise, cognitive biases, or even inherent ambiguity in the annotation task can introduce a large variability in model comparisons [Schoch et al., 2020]. In the context of measuring the reliability in human annotation, it has been shown that selecting a subset of annotators for evaluation may change the performance of models [Van Der Lee et al., 2019, Amidei et al., 2018, Schoch et al., 2020, Amidei et al., 2020], which can be framed as "annotation bias" that also contribute to the benchmark lottery.

## 4 Community bias

Even when viewed as a random process, the task selection bias described in Section 3 alone is sufficient for creating arbitrary selection pressures for machine learning models. We argue however that there is also a higher-level process in which the broader research community influences the task selection, and that counterintuitively leads to the lottery forces not being diminished, but instead more pronounced. This section takes a people perspective of the benchmark lottery and postulates that it is not only the "gamemasters" (benchmark proposers) but also the community that contribute to and reinforce it.

While researchers technically have the freedom to select any dataset to showcase their method, this choice is often moderated by the community. A common feedback in the review process of scientific publications that any ML researcher will face eventually is a criticism of the choice of benchmark. For example *"the method was not evaluated on X or Y dataset"* or *"the method's performance is not SOTA on dataset Z"*. Over time, ML researchers tend to gravitate to safe choices of tasks and benchmarks. For example, most papers proposing new pretrained language models [Lan et al., 2019, Liu et al., 2019, Clark et al., 2020, Yang et al., 2020] evaluate on GLUE even if alternatives exist (see example below for further substantiation). In other words, the selection of tasks commonly used in publication is largely driven by the community. Moreover, whether a benchmark is selected as the canonical testbed or not, is not necessarily governed by the quality of the test examples, metrics, evaluation paradigm, or even what the benchmark truly measures. In fact, an argument that the community is solely responsible for the task selection bias is not without merit, since the community is the final endorser and enforcer of these circumstances. There can be no task selection bias if there is no one to act upon it. To this end, the community might *'double down'* on a benchmark where it becomes almost an unspoken rule for one to evaluate on a particular benchmark. Once a benchmark builds up a following and becomes well-established, it is not hard to imagine that reviewers would ask for results on these benchmarks, potentially regardless of suitability and/or appropriateness. This makes it difficult to fix potentially broken benchmarks.

As foreshadowed, commonly used benchmarks are not immune to containing errors. While these errors are likely to be small (as otherwise they would presumably be noticed early on), they do matter in close calls between competing methods. Northcutt et al. [2021] identified label errors in test sets of 10 of the most commonly-used computer vision, natural language, and audio datasets; for example, there are label errors in 6% of the examples in the ImageNet validation set. They showed that correcting label errors in these benchmarks changes model ranking, especially for models that had similar performance. In the field of NLP, it was later found in SNLI [Bowman et al., 2015], which is a dataset for natural language inference (NLI), a large amount of annotation artifacts exists, and it is possible to simply infer the correct label by only using the premise and not the hypothesis [Gururangan et al., 2018]. It is worth noting that SNLI, being the canonical benchmark for NLI, was easily perceived as mandatory for almost any NLI based research.

The possibility of having such an issue is not only restricted to the peer review process, but it may extend to the public perception of papers after they are published regardless of whether they went through the peer review process or not. The community bias problem can be raised as the community collectively assigning a weighted impact score for doing well on arbitrarily selected tasks. Achieving state of the art on task $Y$ is then deemed significantly less meaningful than doing that for task $X$. Moreover, this is not necessarily done without any explicit reasoning as to why one task is preferred to the other, or even how such a "decision" was made. The main concern with respect to the community bias is that research is becoming too incremental and biased toward the common expectations, since a completely new approach will initially have a hard time competing against established and carefully fine-tuned models. For more discussion and concrete examples on the community bias check Appendix C.

## 5  Benchmarks are stateful

With leaderboards and the continuous publication of new methods, it is clear that benchmarks are stateful entities. At any point in time, the attempt of a new idea for beating a particular benchmark depends on the information gathered from previous submissions and publications. This is a natural way of making progress on a given problem. But when viewed from the perspective of the selective pressures it causes, it creates another kind of lottery. For many machine learning benchmarks, researchers have full access to the holdout set. Although not explicitly, this typically leads to the violation of the most basic datum of "one should not train on test/holdout set" by getting inspiration from already published works by others who presumably report only the best of the numerous models they evaluated on the test set.

Beyond that, it is common to copy-paste hyper-parameters, use the same code, and more recently to even start from pre-retrained checkpoints of previous successful models [2]. In such setups, where the discovery of new models is built on top of thousands of queries, direct or indirect, to the test set, the error rate on test data does not necessarily reflect the true population error [Arora and Zhang, 2021, Blum and Hardt, 2015, Dwork et al., 2015]. The adaptive data analysis framework [Dwork et al., 2015] provides evaluation mechanisms with guaranteed upper bounds on the difference between average error on the test examples and the expected error on the full distribution (population error rates). Based on this framework, if the test set has size $N$, and the designer of a new model can see the error of the first $i-1$ models on the test set before designing the $i$-th model, one can ensure the accuracy of the $i$-th model on the test set is as high as $\Omega(\sqrt{i/N})$ by using the boosting attack [Blum and Hardt, 2015]. In other words, Dwork et al. [2015] state that once we have $i \gg N$ the results on the test set are no longer an indication of model quality. It has been argued that what matters is not only the number of times that a test set has been accessed as stated by adaptive data analysis, but also how it is accessed. Some empirical studies on some popular datasets [Recht et al., 2018, Yadav and Bottou, 2019, Recht et al., 2019] demonstrated that overfitting to holdout data is less of a concern than reasoning from what has been suggested in [Blum and Hardt, 2015]. Roelofs et al. [2019] also studied the holdout reuse by analyzing data from machine learning competitions on the Kaggle and show no significant adaptive overfitting on the classification competitions. Other studies showed that additional factors may prevent adaptive overfitting to happen in practice. For instance, [Feldman et al., 2019b,a] show that in multi-class classification, the large number of classes makes it substantially harder to overfit due to test set reuse. In a recent study, Arora and Zhang [2021] argue that empirical studies that are based on creating or using new test sets (e.g. [Recht et al., 2018, Yadav and Bottou, 2019, Recht et al., 2019]), although reassuring in some level, are not always possible especially in datasets concerning rare or one-time phenomena. They emphasize the need for computing an effective upper bound for the difference between the test and population errors. They propose an upper bound using the description length of models that is based on the knowledge available to model designers before and after the creation of a test set.

From the benchmark lottery point of view, the most important aspect of the above phenomena is that the development of new models is shaped by the knowledge of the test errors of all models before it. First of all, there had been events in the past where accessing the test set more than others, intentionally, secured a margin for victory in the race [3]. In other words, having the ability to access the test set more than others can be interpreted as buying more lottery tickets. Besides, even when there is no explicit intention, the tempting short-term rewards of incremental research polarize people and reinforce the echo chamber effect - leading models are quickly adapted by re-using their code, pre-trained weights, and hyper-parameters are re-used to build something on top of them even faster. Unfortunately, this process makes no time for considering how it affects the statistical validity of results reported on the benchmark.

Another aspect of benchmarks being stateful is that participating in shared tasks at a later stage is vastly different from the time of its inception. By then the landscape of research with respect to the specific benchmark is filled with tricks, complicated and specialized strategies, and know-how for obtaining top performance on the task. The adapted recipes for scoring high are not necessarily universal and may be applicable only to a single narrow task or setup. For example, a publication might discover that a niche twist to the loss function produces substantially better results on the task. It is common for all papers subsequently to follow suit. As an example, the community realized that pre-training on MNLI is necessary for obtaining strong performance on RTE and STS datasets [Liu et al., 2019, Clark et al., 2020], and this became common practice later on. Experience shows that it is not uncommon for benchmark

---

[2]This is in particular common when a paper provides results based on large scale experiments that are not necessarily feasible to redo for many researchers. For instance, the majority of the papers that propose follow up ideas to Vision Transformer [Dosovitskiy et al., 2020] start by initializing weights from the released pretrained models and follow the setups of the original paper. Similarly, several NLP papers use BERT pretrained models and the same hyper-parameters as BERT in their experimental setup.

[3]`https://image-net.org/challenges/LSVRC/announcement-June-2-2015`

tasks to accumulate lists of best practices and tricks that are dataset- and task-specific [4]. Whether a novel algorithm is able to make use of these tricks (or whether they are available at all) is again a form of lottery, in which models that cannot incorporate *any* of the earlier tricks are significantly disadvantaged.

# 6 Rigging the lottery: making your own luck

For some tasks and problems, there are already standard benchmarks and established setups that are followed by most of the community. However, for some others, inconsistencies in the employed benchmarks or reported metrics can be observed. This diversity of evaluation paradigms makes comparisons between publications extremely difficult. Alternatively, in some cases, there is simply no standard benchmark or setup, either because the problem is still young, or because there has never been an effort to unify the evaluation. Sometimes this is due to the high computational cost of proper evaluation, like when reporting variance over multiple random seeds is important [Bouthillier et al., 2019]. While in other instances, the root cause is of behavioral nature, where researchers prefer to showcase only what their method shines at - oftentimes to avoid negative reviews, unsuccessful experiments, although performed, are simply not reported. Here, we study two known examples of this issue, which we refer to as *rigging the lottery*.

## 6.1 Recommender systems and benchmark inconsistencies

Unlike the fields of NLP or CV, there are no well-established evaluation setups for recommender systems [Zhang et al., 2019] that provide canonical ranked lists of model performance. While there has been a famous Netflix prize challenge[5], this dataset has not been extensively used in academic research or for benchmarking new models. Moreover, even popular datasets like MovieLens [Harper and Konstan, 2015] or Amazon Reviews [He and McAuley, 2016] generally do not have a canonical test split, metric or evaluation method. Therefore, it is still quite unclear about which modern RecSys method one should adopt, as model comparisons are difficult to interpret [Dacrema et al., 2019].

Furthermore, RecSys evaluation is also very challenging for a number of reasons. (*i*) Different recommendation platforms tackle slightly different problems (e.g retrieval [Yi et al., 2019], ranking ([Pei et al., 2019]), or multitask learning ([Zhao et al., 2019])), and each requires their own evaluation setup. (*ii*) As is common for user interacting systems, user's reaction towards different algorithms can be different. Constructing offline datasets of user behaviors from an existing system creates an off-policy evaluating challenge [Swaminathan et al., 2016]. (*iii*) A real-world recommendation system trains on billions of users and items, the scale of user-item interactions makes it extremely difficult to create a complete dataset containing all possible user-item interactions [He et al., 2016b]. As a result, evaluation setups in many recommender system papers tend to be arbitrary.

There exists a small number of public datasets (see Appendix E), such as MovieLens [Harper and Konstan, 2015] or Amazon Product Review [He and McAuley, 2016] that are commonly used for evaluating recommender systems. However, even these datasets are tweaked differently in various publications, leading sometimes to contradictory results [Rendle et al., 2020, Zheng et al., 2019]. For example, some papers use Hit Ratio and NDCG as evaluation metrics [He et al., 2017], while others resort to using Recall@K [Zheng et al., 2019]. Interestingly, in this particular example, the same methods reverse their performance when a different metric is used. Holdout test sets can also be created differently, with some papers for example using random split [Beutel et al., 2017] and others using an out-of-time split [Zhang et al., 2020].

While the majority of this paper discusses cases where a standardized benchmark may lead to biased progress in the ML community, here we instead discuss the *exact opposite* - implications of having no consensus datasets or evaluation setups. Having no unified benchmark for the community to make progress on has numerous flaws. To name a few, (*i*) this hinders progress in the field, while possibly (*ii*) creating an illusion of progress. It is not surprising that under these circumstances researchers (potentially unknowingly) tend to find good experimental setups that fit their models. For a case study on inconsistencies of the evaluation setup in ALE benchmark check Appendix D.

# 7 What can we do?

While the previous sections of the paper focused on the challenges that arise from the lottery-like interaction between ML benchmarks and the research community, here we would like to show that there are reasons to be optimistic about future developments in this regard. We present suggestions for improving the idea benchmarking process in ways that make it less of a lottery. These recommendations can be also

---

[4]As an example, for achieving scores that are comparable to top-ranked models on the GLUE benchmark, there are a series of extremely specific actions and setups used in pretraining/finetuning that are known as "standard GLUE tricks" introduced/used by submissions to the leaderboard [Liu et al., 2019, Yang et al., 2019, Lan et al., 2019]. Check the Pre-training and fine-tuning details in the appendix of [Clark et al., 2020].

[5]https://netflixprize.com/index.html

framed as checklists[6] for different parts of the process, like making benchmarks, using benchmarks, evaluation of a new ideas. Appendix F presents a proposed benchmarking checklist for the review process.

## 7.1 Investing in making guidelines

We believe that the risks of "rigging the lottery" that is described in Section 6 can be minimized by standardizing the recipe for creating and using benchmarks.

**Guidelines for creating benchmarks.** Investing into shared guidelines for creating new benchmarks can be extremely beneficial to the long-term health of the research community. In our view, such guidelines should include the current best practices and aspects that require special attention; and should highlight potential concerns for issues that may emerge in the future when different models and algorithms are applied to the benchmarks. Fortunately, there have been some efforts in providing guidelines and best practices for making new benchmarks. For example, Zhang [2021] discusses the need for how robotic warehouse picking benchmarks should be designed. Kiela et al. [2021] proposed a framework for benchmarking in NLP that sets clear standards for making new tasks and benchmarks. Denton et al. [2020] look at the dataset construction process with respect to the concerns along the ethical and political dimensions of what has been taken for granted, and discuss how thinking about data within a dataset must be holistic, future-looking, and aligned with ethical principles and values. Bender and Friedman [2018] also proposed using data statements for NLP datasets in order to provide context that allows users to better understand how experimental results on that dataset might generalize, how software might be appropriately deployed, and what biases might be reflected in systems built on the software.

In Section 6 we pointed out that sometimes the blocking factor or conduction rigorous evaluation is the high computational costs, in particular for the academic environment. For instance, analyzing all possible sources of variance in the performance is prohibitively expensive. As a potential solution to this problem, the community can invest more in setting up initiatives like reproducibility challenges[7] or specialized tracks at conferences that offer help in terms of expertise, infrastructure, and computational resources for extensive evaluation to the papers submitted to that conference.

**Guidelines for benchmark usage.** Besides the necessity of making guidelines for "how to make new benchmark", it is important to have clear guidelines for "how to use a benchmark", which for instance includes the exact setup that the benchmark should be used for evaluation or how the results should be reported. This would be a great help with reducing the instances of rigging the lottery prevalent in some domains (Section 6). There are also several efforts targeting this goal. For instance, Albrecht et al. [2015], Machado et al. [2018] propose specific standards for the ALE benchmark (discussed in Section D.1). Ethayarajh and Jurafsky [2020] argue against ranking models merely based on their performance and propose to always report *model size*, *energy efficiency*, *inference latency*, and metrics indicating model *robustness* and *generalization to the out-of-distribution data*. Gebru et al. [2018] proposed that every dataset be accompanied by a datasheet that documents its motivation, composition, collection process, recommended uses, etc with the goal of increasing transparency and accountability, mitigating unwanted biases in ML systems, facilitating greater reproducibility, and helping researchers and practitioners select more appropriate datasets for their chosen tasks.

Another important problem that can benefit from established regulation is the hyper-parameter tuning budget used by researchers to improve their model performance. Spending enough time and compute to precisely tune hyper-parameters of the model or the training process can improve the results a great deal [Li et al., 2018, Bello et al., 2021, Steiner et al., 2021]. Given that, a guideline on limiting the budget for the hyper-parameter tuning can curb the improvements that are solely based on exhausting hyper-parameter search and gives a chance to have comparisons that are tied less to the computational budget of the proposing entity, but more to the merits of the methods themselves.

**Guidelines for conferences and reviewers.** There have been attempts to ameliorate the problems related to the benchmark lottery, especially its community biases and the statefulness aspects (Sections 4 and 5). For example, NLP conferences have specially called out *"not being SOTA"* as an invalid basis for paper rejection[8] We believe it is possible to leverage education through the review process in order to alleviate many negative aspects of benchmark lottery.

As an example, we can make sure that in the review process, scores on a particular benchmark are not used for immediate comparison with the top-ranking method on that benchmark, but rather as a sanity check for new models and simply an efficient way of comparing against multiple baselines. This way, fundamentally new approaches will have a chance to develop and mature instead of being forced to compete for top performance right away or get rejected if not succeeded in the early attempts.

---

[6]Similar to the reproducibility checklist `https://www.cs.mcgill.ca/~jpineau/ReproducibilityChecklist.pdf` [Dodge et al., 2019]

[7]For instance `https://paperswithcode.com/rc2020`, `https://reproducibility-challenge.github.io/iclr_2019/`, or `https://reproducibility-challenge.github.io/neurips2019/`

[8]`https://2020.emnlp.org/blog/2020-05-17-write-good-reviews`.

## 7.2 Statistical significance testing

The presence of established benchmarks and metrics alone does not necessarily lead to a steady improvement of research ideas; it should also be accompanied by rigorous procedures for comparing these ideas on the said benchmarks. For example, Armstrong et al. [2009] discuss the importance of comparing improvements to the strongest available baselines, however, the question of how do we know that if a new model $B$ is *significantly* better than its predecessor model $A$ remains anything but solved 10 years later [Lin et al., 2021].

**Benchmark results as random samples.** Machine learning models are usually trained on a training set and evaluated on the corresponding held out test set, where some performance metric $m$ is computed. Because model training is subject to sources of uncontrolled variance, the resulting metric $m$ should be viewed as a single sample from the distribution describing the model's performance. Because of that, deciding which of the two models is better based on point estimates of their performances $m_A$ and $m_B$ may be unreliable due to chance alone. Instead distributions of these metrics $p(m_A)$ and $p(m_B)$ can be compared using statistical significance testing to determine whether the chance that model $A$ is at least as good as model $B$ is low, i.e. $p(A \leq B) < \alpha$ for some *a priori* chosen significance level $\alpha$. Estimation of $p(A \leq B)$ forms for the crux of statistical significance testing. It can be done either by using parametric tests that make assumptions on distributions $p(m_A)$ and $p(m_B)$ and thus often need fewer samples from these distributions, or by using non-parametric tests that rely on directly estimating the metric distributions and require more samples.

The popularity of standardized benchmarks and exponential growth in the amount of research that the ML community has experienced in recent years[9] exacerbate the risk of inadvertently misguiding research through lax standards on declaring a model as an improvement on the SOTA. Indeed, if point estimates are used in place of statistical significance testing procedures, sampling $m'_A \sim p(m_A)$ and $m'_B \sim p(m_B)$ such that $m'_B > m'_A$ is only a matter of time, even if performance of the two models is not actually different. Note that this is *not* the same as the issue described in Section 5, but could instead be thought of as winning a lottery if you purchase enough lottery tickets.

**Beyond a single train-test split.** Unfortunately, researchers rarely go through the process of collecting strong empirical evidence that model $B$ significantly outperforms model $A$. This is not surprising. As discussed in Bouthillier et al. [2021], obtaining such evidence amounts to running multiple trials of hyper-parameter optimization over sources of variation such as dataset splits, data ordering, data augmentation, stochastic regularisation (e.g. dropout), and random initialization to understand the models' variance, and is prohibitively expensive[10]. If studied at all, mean model performance across several random parameter initializations is used for declaring that the proposed model is a significant improvement. This is vastly sub-optimal because dataset split contributes the most to model variance compared to other sources of variation [Bouthillier et al., 2021]. However, providing multiple dataset splits to estimate this variance is not standard practice in benchmark design.

Benchmarks typically come with a single fixed test set, and thus could even be said to unintentionally discourage the use of accurate statistical testing procedures. This is particularly problematic for mature benchmarks, where the magnitude of model improvements may become comparable to the model variance. Systematic variance underestimation may lead to a series of false positives (i.e. incorrectly declaring a model to be a significant improvement) that stall research progress, or worse - lead the research community astray by innovating on "improved overfitting" in place of algorithmic improvements. Going forward, one way of addressing this limitation is to design benchmarks with *multiple* fixed dataset splits. As an added benefit, model performance reported across such standardized splits would also enable the application of a variety of statistical tests not only within the same study, but also across publications.

**Benchmark design with statistical testing in mind.** The choice of a suitable statistical testing procedure is non-trivial. It must consider the distribution of the metric $m$ that is being compared, the assumption that can be safely made about the distribution (i.e whether a parametric test is applicable or a non-parametric test should be used), the number of statistical tests performed (i.e. whether multiple testing correction is employed) and can also change as the understanding of the metric evolves [Demšar, 2006, Bouthillier et al., 2021, Lin et al., 2021]. We, therefore, recommend that benchmark design is accompanied by the recommendation of the suitable statistical testing procedures, including the number dataset splits discussed above, number of replicates experiments, known sources of variance that should be randomized, the statistic to be computed across these experiments and the significance level that should be used for determining statistically significant results. This would not only help the adoption of statistical testing for ML benchmarks, but also serve as a centralized source for best

---

[9]https://neuripsconf.medium.com/what-we-learned-from-neurips-2020-reviewing-process-e24549eea38f

[10]Although Bouthillier et al. [2021] also propose a pragmatic alternative to the exhaustive study of all source of variation.

practices that are allowed to evolve. A detailed discussion of statistical testing is outside of the scope of this paper, and we refer interested readers to [Bouthillier et al., 2021, Dror et al., 2017] for an overview of statistical testing procedures for ML.

**Beyond a single dataset.** Often we are interested in understanding whether model $B$ is significantly better than model $A$ *across a range of tasks*. These kinds of comparisons are facilitated by benchmarks that span multiple datasets (e.g. VTAB or GLUE). Already the question of what it means to do better on a multi-task benchmark is non-trivial due to the task selection bias (see Section 3) - is it sufficient for model $B$ to do better on average; or should it outperform model $A$ on all tasks? It is not surprising that the statistical testing procedures for such benchmarks are also more nuanced - the answer to this question leads to different procedures. It is unclear whether the average metric across datasets, a popular choice for reporting model performance, is meaningful[11] because the errors on different datasets may not be commensurable, and because models can have vastly different performance and variances across these datasets. For this reason, more elaborate procedures are required. For example, for the case when we are interested in seeing whether $B$ outperforms $A$ on average Demšar [2006] propose to ignore the variance on individual datasets and treat the model $A$ and $B$'s performance across datasets as samples from two distributions that should be compared. They recommend that the Wilcoxon signed-rank should be used in such a setup; but the recommended can have limited statistical power when the number of datasets in the benchmark is small. Alternatively, for cases when we are interested in seeing whether $B$ is better than $A$ on all datasets Dror et al. [2017] propose to perform statistical testing on each of the datasets separately while performing multiple testing corrections. Here again the "right" statistical testing procedure depends on the benchmark, its composition, and the criteria for preferring one model over another; and we believe that the community would benefit if these questions were explicitly answered during benchmark design.

## 7.3 Rise of living benchmarks

Another major issue for many popular benchmarks is "creeping overfitting", as algorithms over time become too adapted to the dataset, essentially memorizing all its idiosyncrasies, and losing the ability to generalize. This is essentially related to the statefulness of benchmarks discussed in Section 5. Besides that, measuring progress can be sometimes chasing a moving target since the meaning of progress might change as the research landscape evolves. This problem can be greatly alleviated by for instance changing the dataset that is used for evaluation regularly, as it is done by many annual competitions or reoccurring evaluation venues, like WMT[12] or TREC[13]. Besides that, withholding the test set and limiting the number of times a method can query the test set for evaluation on it can also potentially reduce the effect of adaptive overfitting and benchmark reuse. In a more general term, an effective approach is to turn our benchmarks into "living entities". If a benchmark constantly evolves, for instance, adds new examples, adds new tasks, deprecates older data, and fixes labeling mistakes, it is less prone to "tricks" and highly robust models would find themselves consistently doing well across versions of the benchmark. As examples of a benchmark with such a dynamic nature, GEM is a living benchmark for natural language generation [Gehrmann et al., 2021] or Dynabench [Kiela et al., 2021] proposes putting humans and models in the data collection loop where we continuously reevaluate the problem that we really care about.

## 8 Epilogue

Ubiquitous access to benchmarks and datasets has been responsible for much of the recent progress in machine learning. We are observing the constant emergence of new benchmarks. And on the one hand, the development of benchmarks is perhaps a sign of continued progress, but on the other hand, there is a danger of getting stuck in a vicious cycle of investing in making static benchmarks that soon will be rejected due to the inflexible flaws in their setup, or lack of generality and possibility for expansion and improvements. We are in the midst of a data revolution and have an opportunity to make faster progress towards the grand goals of artificial intelligence if we understand the pitfalls of the current state of benchmarking in machine learning. The "benchmark lottery" provides just one of the narratives of struggling against benchmark-induced model selection bias. Several topics we touched upon in this paper are discussed in the form of opinions or with a minimum depth as a call for further discussion. We believe each subtopic deserves a dedicated study, like how to better integrate checks for ethical concerns in the mainstream evaluation of every existing benchmark, how to develop tools and libraries that facilitate the rigorous testing of the claimed improvements, or a deep investigation of the social dynamics of the review process and how to improve it. In the end, there are many reasons to be excited about the future - the community is continuously taking positive delta changes that contribute to fixing issues with measuring progress in the empirical machine learning.

---

[11] In fact for that reason it was not a popular choice until recently [Demšar, 2006].

[12] http://statmt.org/

[13] https://trec.nist.gov/

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
