# OpenReview forum: "The Benchmark Lottery"
_NeurIPS.cc/2021/Track/Datasets_and_Benchmarks/Round1 — Submitted to NeurIPS 2021 Datasets and Benchmarks Track (Round 1)_

### Official Review · Reviewer_J25f · 2021-07-01
**Interesting and beneficial paper; I hope the authors clean it up during the rebuttal**

**Rating:** 5
**Confidence:** 4
**Correctness:** See weakness (1).

**Strengths:**

Update on 19 July 2021:
I thank the authors very much for making many changes to the paper just before the discussion period ended. I think the paper has now improved. Unfortunately, there are still some signs that the paper could be polished further. For example, some of the new sections seem a bit rushed and could be thought through more, and there were a couple things the authors promised but haven't done yet. I think this paper has the potential to be a very strong paper in the next round. I will keep my rating as it is.

Original review:

This paper is very interesting and has a lot of great ideas and concepts. I think this is an important and impactful paper. I can see this paper as a stepping stone in driving the field to have much better benchmarking practices (and not be obsessed with certain benchmarks) and also to conduct better reviews (e.g., not rejecting based on the results on a small number of benchmarks). Better benchmarking and better reviewing are both critical to the health of machine learning research, so this paper has a large potential impact.

This paper is a great resource for all types of pitfalls in benchmarking practices, and a good resource for mitigating these pitfalls.

**Weaknesses:**

(1) In many sections, the authors make claims that are not tested or cited. This makes it feel more like an opinion piece rather than scientific research. I find these claims agreeable, but the authors must find a way to justify them, or else take them out of the paper. Here is a non-exhaustive list of examples:
(1.1) Section 4: “reviewers will ask for results on common benchmarks potentially regardless of the suitability and/or appropriateness.”
(1.2) Section 4: “[pressure to provide strong results on common benchmarks] also extends to the public perception of papers”.
(1.3) Section 5: “it is common to copy-paste hyper-parameters ... even start from pre-retrained checkpoints of previous successful models”
(1.4) Section 6: “Experience shows that it is not uncommon for benchmark tasks to accumulate lists of best practices and tricks that are dataset- and task-specific”
(1.5) Section 6: “Many of the adapted recipes for scoring high are non-universal.”
(1.6) Section 6.1: “model comparisons are currently all over the place.”
(1.7) Appendix B “>99% evaluate on GLUE”

(2) The overall structure of the writing is great, but the low level writing is sometimes not good. See a list of grammatical errors and comments in the final section of this review.

(3) In Section 3, in the SuperGlue experiments, all of the 55 models still have the transformer paradigm. In the spirit of the benchmark lottery, wouldn’t this experiment be stronger if there was a more diverse list? For the next experiment on VTAB, the choice of models is not even discussed in the main body. The choice of models significantly affects the experiments, so these details should be included in the main paper. Also, 0.7-0.9 Kendall Tau does not seem that bad. Can the authors give an explanation for this?

(4) The authors replied “Yes” to checklist question 1.b, but I do not see a discussion of limitations. Can the authors let me know if I missed it somewhere? As a related question, are there any tasks where the benchmark lottery does not have a large effect? Perhaps VTAB, since the Kendall Tau scores were relatively high.

(5) This paper had experiments, but did not release the code for these experiments, therefore, the experiments have low reproducibility. The paper’s main contributions are new ideas and concepts, but experiments were still performed to test some of those ideas, and so it would improve the paper to make those experiments reproducible. The call for papers suggests using a reproducibility framework.

I really enjoyed this paper, but it is a bit too rough to accept in the current form. I currently give this paper a weak reject, but I am willing to increase my rating substantially if my concerns are partially or fully addressed during the review process. For example, the authors could provide a link to an updated pdf.

**Additional Feedback:**

Grammatical errors and minor comments

- Page 1, para 2, sentence 1: “assumptions and algorithmic” -> “assumptions, and algorithmic”
- Page 2, para 2, “On a whole” -> “As a whole”
- Page 2, bullet 5, “lack of a well-established” -> “lack of well-established”
- Page 3, last sentence of section 2: fix “we diving into”
- Page 4, second to last section, fix “in the case VTAB” and “training data for downstream is”
- Page 4, para “One of the main difficulties...” I don’t see the point of this paragraph. It should be connected better with the rest of section 3.2, or removed.

- Page 5, para 1 “counter intuitively” is one word
- Page 5, para 2: “exist (See” -> “exist (see”
- Page 5, para 2: “irregardless” -> “regardless”
- Page 5, para 2: SNLI is defined in the last section, but it should be defined the first time it is mentioned, in the previous sentence.
- Sec 5, sentence 1: “as a model attempt ... depend on the information...” What does this mean? Is there a typo there?
- Page 6, para 1: fix “from what have suggested in”
- Page 6, end of sec 5: “events in the past that” -> “events in the past where”
- Page 6, end of sec 5: “As an example, when the community realized...” -> “As an example, the community realized...”
- Page 6, section 6: “which we refer as rigging...” -> “which we refer to as rigging...”
- Page 7, first sentence “this dataset or task have” -> “this dataset has”
- Page 7, “that each requires” -> “and each requires”
- End of section 6: “we instead discusses” -> “we instead discuss”
Several more grammatical errors in section 7. I will stop listing each one individually.

- Page 8, footnote 5: split the link into two lines
- Page 22: “potentially missed.)” -> “potentially missed)”
- Page 23: “number distinct” -> “number of distinct”

- Figure 1: only the diamond is labeled as Top-10. What values of k are the other ones? From the text, it sounds like square is top-1? What is circle, top 55? How come the text specifies four values but there are five ticks?

- It seems hat top-1 here is most important, since the highest performing models are what drives the field. But the top-1 has low disagreement on all subsets >1, so isn’t this evidence against the benchmark lottery?

- Appendix Figure 3 is nice, but it is much too large to fit on a page. I suggest having a much smaller representative subset as Appendix Figure 3, and put the entire diagram available online (e.g. if there is a Github repo for this project). Also, there is no legend for the colors.

**Clarity:**

The overall structure of the paper is sound, but certain sections could have clarity improved. See “Additional Feedback.”

**Documentation:**

See weakness (5).

**Ethics:**

There are no ethical concerns that warrant further review.

**Relation To Prior Work:**

Although there is no section on related work, each section sufficiently discusses many prior papers which give similar ideas or analyses. For example, Section 5 mentions “A Meta-Analysis of Overfitting in Machine Learning.”

**Summary And Contributions:**

The authors introduce the notion of the benchmark lottery. This is the idea that the ML community conforms to certain benchmarks, and therefore much research effort goes into improving certain algorithm classes which have strong performance on these particular benchmarks. The paper makes the case for the overall fragility of ML benchmarking, and gives suggestions to mitigate these issues.

The paper specifically makes points about the life-cycle of a benchmark, task selection bias, community bias, “stateful entities” of benchmarks, “rigging the lottery”, and recommendations to mitigate these issues.

---

> ### Author Response · Authors · 2021-07-15
> **Response to reviewer J25f**
>
> Thank you for your suggestions  and extensive comments. We have updated our submission to incorporate the feedback in the comments.
> We have fixed the issues in the writing including grammatical errors and ambiguous sentences. We have also improved the plots (fixed the issue with rendering the legends in plots). We will also open-source the code/colab for the posthoc analyses we have done in this paper for our case studies in a github repo along with the link to downloadable high resolution / zoomable version of all plots (and replace some of the plots in the paper with a summarized version). We would also like to thank the reviewer for the comment on the need for justifications for some of the statements in the paper. We have edited the text to add references or rephrase statements in order to make it more clear that some of those statements are speculations and opinions based on general observations or limited data (e.g. checking a sample of 100 papers rejected from ICLR 2019/2020 due to some issue with the benchmark they used/didn’t use). Besides the modifications we made on the revised paper to address all the comments and raised issues, we respond to some of the comments/suggestions that require more explanation:
>
> > In Section 3, in the SuperGlue experiments, all of the 55 models still have the transformer paradigm. In the spirit of the benchmark lottery, wouldn’t this experiment be stronger if there was a more diverse list?
>
> The 55 models that we cover for the SuperGLUE experiments are the best performing (each got SOTA when they were proposed) models and although a large portion of the list comprises of Transformer-based model (due to hype in the community as well as transformer’s success on various setups), the list also covers non-transformer models, like lightweight convolutions and Dynamic convolutions. We have corrected the text to reflect this point.
>
> > For the next experiment on VTAB, the choice of models is not even discussed in the main body. The choice of models significantly affects the experiments, so these details should be included in the main paper. Also, 0.7-0.9 Kendall Tau does not seem that bad. Can the authors give an explanation for this?
>
> We have added explanations on the models we cover in our experiments in the main body (which were pushed to the appendix initially due to space limitations). For VTAB, the rank correlation between the pre-defined “subcategories” is not too low as you pointed out, however: (1) the correlation between “tasks” and the full VTAB score can be even negative as it’s shown in Figure 2-b. (2) more importantly, as we shown in Appendix “VTAB: Agreement on top-ranked models across sub-categories and tasks” there is a “full disagreement” between the subsets of subcategories in terms of the top-1 (winning) method. In general, the appendix shows that for varying subsets of tasks or even subcategories, we have several different models showing the best performance. Given that most of the time, the top-k models with k=1, 2, or 3 become the main candidates for further developments, with such a disagreement in that region of the rank list, we can see the effect of the benchmark lottery kicks in where a single score introduces a model as the winner, where subsets of tasks or categories disagree on that.
>
> > The authors replied “Yes” to checklist question 1.b, but I do not see a discussion of limitations. Can the authors let me know if I missed it somewhere? As a related question, are there any tasks where the benchmark lottery does not have a large effect? Perhaps VTAB, since the Kendall Tau scores were relatively high.
>
> As the main limitation of this paper, several topics we touched upon are discussed in the form of opinions or with a minimum depth as a call for further discussion. We believe each subtopic deserves a dedicated study, like how to better integrate checks for ethical concerns in the mainstream evaluation of every existing benchmark, how to develop tools and libraries that facilitate the rigorous testing of the claimed improvements, or a deep investigation of the social dynamics of the review process and how to improve it. We have added this explanation to the paper to make this point more explicit.
> Also regarding the question “Are there any tasks where the benchmark lottery does not have a large effect?”; for all the case studies we covered in the paper, we believe there is, to different extents, a lottery effect that contributes to the process of introducing a single model as a winner. Such a phenomenon, i.e. the benchmark lottery, is also not just the product of the way a benchmark is designed, but also many different factors, like the community bias, the pace of progress and the frequency of emerging new solutions, etc., that are not easy to quantify, making it difficult to explicitly identify a particular observation as a result of the benchmark lottery or claim “no benchmark lottery” in a specific setup.

---

> > ### Author Response · Authors · 2021-07-15
> > **Response to reviewer J25f - Part 2**
> >
> >
> > > This paper had experiments, but did not release the code for these experiments, therefore, the experiments have low reproducibility. The paper’s main contributions are new ideas and concepts, but experiments were still performed to test some of those ideas, and so it would improve the paper to make those experiments reproducible. The call for papers suggests using a reproducibility framework.
> >
> > For the analysis and case studies in this paper, we didn’t do any experiments in terms of implementing, training or evaluating different methods given different benchmarks. All the analyses are posthoc processing of the tabular data containing scores of different methods on different tasks/benchmarks that come from publicly available leaderboards and published papers (provided citation/links in the paper). We will share the colab that runs the posthoc analyses in a github repo.
> >
> > >It seems hat top-1 here is most important, since the highest performing models are what drives the field. But the top-1 has low disagreement on all subsets >1, so isn’t this evidence against the benchmark lottery?
> >
> > When looking at the top-scoring models (winners), for subsets of size 1 (individual tasks), we have 6 different winners. When looking at the subsets of sizes 2, 3, and 4, we still have 6, 8, and 6 different winners, respectively. Although the portion (6/28, 8/56, and 6/70) becomes small, the fact that the absolute number of top-ranked models in various scenarios is still 6 (or 8) shows that taking one of them as the winner (based on the average score) can still be just winning a lottery and the chance of all these 6 (or 8) models getting the same amount of attention by the community is extremely low, so we believe that the benchmark lottery argument holds valid.

---

### Official Review · Reviewer_uEg4 · 2021-07-04
**Paper that surveys the problems with the current benchmarking process well but stops short of providing enough concrete suggestions**

**Rating:** 5
**Confidence:** 4

**Strengths:**

The paper does a good job of surveying several papers that point out flaws in the current benchmarking process. I feel such introspection is really needed and valuable in the community. Many of the suggestions in the paper should lead to better benchmarking and are widely applicable to the community.

**Weaknesses:**

While the paper does a good job of pointing out the current problems with benchmarking, I think it doesn't make enough concrete suggestions to overcome these. Firstly, I would encourage the authors to make a "Benchmark Checklist" similar to the Reproducibility Checklist https://www.cs.mcgill.ca/~jpineau/ReproducibilityChecklist.pdf. This could lead to greater adoption because the authors would distill the takeaways from the many papers which would be a great value-add to the community.

One key aspect which I think is not touched upon enough is the hyperparameter (HP) tuning budget used by researchers. Given the large variance of the models used by the community today, given enough time to tune hyperparameters, any method with enough variance can, in principle solve a task as far as I understand (this is related to Section 5). So, a guideline on limiting HP tuning budget should be added I feel. This should help curb "throwing compute" at a problem to "solve" it.

Another suggestion from my side would be that authors of methods be required to submit code and reviewers be required to test it on a benchmark of their choice. This is like having a blind dataset to test a method on. Naturally, this should be low overhead for reviewers and so should be done on reasonably sized datasets - this process could even be automated on "test servers" if the authors conform to an API. This also promotes making the code easier to use. (This should only be used as a sanity-check and not to reject the paper if the method does not achieve SOTA - just as the authors propose for the benchmarks used in the paper itself.)

The above suggestions could help tackle the illusion of progress in the community as seen in the paper: A Metric Learning Reality Check: https://www.ecva.net/papers/eccv_2020/papers_ECCV/papers/123700681.pdf

Another suggestion, which may be more subjective on my part, is that in community-driven conferences only results on public benchmarks should be considered for scoring reviews.


**Additional Feedback:**

In Figure 1, why wasn't rank correlation used? I'm not surprised there is a great mismatch in trying to match exact ranking orders on the sub-tasks. Should we really be trying to match the exact ranking orders?

>Section 2 discusses how benchmarks can influence long-term research directions in a given (sub-)field, and describes the life cycle of a benchmark.

I did not see the life cycle of a benchmark being discussed in this section.

>Kolesnikov et al. [2019] tried different ranking
aggregation methods from [Balduzzi et al., 2018] and showed they were fairly correlated.

I'm not sure what this sentence is doing at the end of the paragraph it is in. It feels a bit like it doesn't belong there. Is it about ranking aggregation on VTAB? Doesn't it go against what the rest of the paragraph is saying?

>Kiela et al. [2021] proposed a framework
that sets clear standards for making new tasks and benchmarks.

I think the sentence by itself is misleading if we don't qualify that the paper is for NLP.

>such as the fact that some tasks use the same input data (e.g CLEVR-Count and CLEVR-Dist), thus upweighting those domains;

Looking at Figure 2, those sub-tasks actually have opposite rank correlations with the full VTAB score. So you might want to also say that upweighting those domains leads to also noisier results.

>And on the one hand, the development of benchmarks is perhaps a sign of continued progress, but on the other hand, there is a danger of getting stuck in a vicious cycle of investing in making benchmark that soon will be rejected due to their flaws.

Isn't this contradicting somewhat the point about constantly updating benchmarks to remove flaws in them?

Repeated references:
1) Improvements that don’t add up: ad-hoc retrieval results since 1998.
2) Significant improvements over the state of the art? a case study of the MS MARCO document ranking leaderboard.


There were no line numbers in the submission, so I assume the style file used for the submission deviated from the one for this track.

The appendix was in the main submission, which as I understood was not allowed.

In the paper checklist, I would like to see the relevant paper sections referenced in the first point.



I have summarised the suggestions proposed by the authors, that I could find, in a list here, in case the authors want to indeed create a checklist as I suggested above:
>As an example, we can make sure that in the review process, scores on a particular benchmark are not used for an immediate comparison with the top-ranking method on that benchmark, but rather as a sanity check for new models and simply an efficient way of comparing against multiple baselines.

>Bouthillier et al. [2021] showed that dataset split contributes the most to model variance compared to other sources of variation. However, providing multiple dataset splits to estimate this variance is not standard practice in benchmark design ... addressing this limitation is to design benchmarks with multiple fixed dataset splits

>benchmark design is accompanied by the recommendation of the suitable statistical testing procedures.

>changing the dataset that is used for evaluation regularly

>benchmark constantly evolves, for instance, adds new examples, adds new tasks, deprecates older data, and fixes labeling mistakes, it is less prone to “tricks” and highly robust models would find themselves consistently doing well across versions of the benchmark






**Clarity:**

The paper is well-written and easy to follow generally. There were some points I did not completely understand. These are mentioned later in the review. There were also several minor grammatical mistakes.

**Correctness:**

I believe the claims are correct.


**Documentation:**

N/A

**Ethics:**

I believe ethical concerns about benchmarks were not addressed in detail and some guidelines regarding paying attention to ethical concerns when designing benchmarks could be added.


**Relation To Prior Work:**

The paper does a good job of surveying several papers. I believe "A Metric Learning Reality Check" and "The Reproducibility Checklist" could be further cited.

**Summary And Contributions:**

The paper reviews how task selection bias and community bias can lead to faulty conclusions of progress in the research being performed. It describes the process as a benchmark lottery. The paper also discusses how re-use of the same established benchmarks can lead to overfitting to the respective test sets. The paper says further that in communities that do not have established benchmarks, the situation is even worse as there are no set evaluation procedures and researchers may fit benchmarks to models. The paper also provide some suggestions to mitigate the problems.

---

> ### Author Response · Authors · 2021-07-15
> **Response to reviewer uEg4**
>
> Thank you for your comments and suggestions.
> We have updated the paper where we fixed some issues pointed out by the reviewer about ambiguous or unconnected sentences and some missing parts that we still have references to (which appeared when shrinking the paper into the page limit).
> We have also expanded some parts, like the effect of negative rank correlation on CLEVR tasks on the aggregate score. We have also cited the missing references pointed out by the reviewer. Last, we addressed points that are raised as “Additional Feedback” by the reviewer by updating related parts and fixed the issues with repeated entries in the reference list as well as the style/format of the submission in the revised version. Bellow, we will discuss some of the questions raised by the reviewer:
>
> > [...] I would encourage the authors to make a "Benchmark Checklist" similar to the Reproducibility Checklist [...]
>
> Thanks for the great suggestion. We have added a “Benchmarking Checklist” section to the appendix that discusses some concrete items related to the benchmarking that can be checked by the reviewers when evaluating the quality of benchmarking in a paper to minimize the benchmark lottery effect. It’s noteworthy that the “Datasheets for Datasets” paper (https://arxiv.org/pdf/1803.09010.pdf) also provides a checklist for benchmark creation that covers items related to the motivation of the benchmark, composition, collection process, and recommended uses.
>
> > One key aspect which I think is not touched upon enough is the hyperparameter (HP) tuning budget used by researchers [...]
>
> As per the reviewer's suggestion, in the revised paper, we briefly discuss how much hyper-parameter tuning can affect the performance of models and talk about the need for a setup that limits the hyper-parameter tuning budget for the sake of having a fair comparison between different methods in a way that is less tied to the computational budget of the proposing entity and more about the merits of the method (thanks for sharing your thoughts on this).
>
>
> > I believe ethical concerns about benchmarks were not addressed in detail and some guidelines regarding paying attention to ethical concerns when designing benchmarks could be added.
>
> We believe there is no ethical concern about the content of our paper itself, but regarding the importance of considering ethical factors in the benchmarking, we strongly agree with the review and related to this in the section where we discuss the necessity of “Investing in making guidelines”, we briefly discuss this issue and referred to two important work in this direction (1) Bringing the people back in: Contesting benchmark machine learning datasets, by Denton et al., and (2) Datasheets for datasets, by Gebru et al., and explained their proposals for how to take into account concerns related to ethics, transparency, accountability, and unwanted biases in datasets creation and usage. We agree with the importance of this topic and we leave more focused study on this specific area for future dedicated research work.
>
>
> > In Figure 1, why wasn't rank correlation used? I'm not surprised there is a great mismatch in trying to match exact ranking orders on the sub-tasks. Should we really be trying to match the exact ranking orders?
>
> This is a great question. Having inconsistency between the exact rank lists is of course expected and not surprising. However, Figure 1 focuses on the top-ranked models, which are those that are most likely to be adapted for future development by researchers or deployed by practitioners. Focusing on the exact top-k ranking, with a small$k$ shows the disagreement on the important region of the rank list. For instance, the fact that in 6 out of 8 individual tasks, we have different models as the winner (top-1) is one of the important points that is meant to be highlighted in Fig1. We have added some explanation to make this point more clear, but for the sake of completeness, we also added the rank correlation plot for the SuperGLUE benchmark to the appendix.

---

### Official Review · Reviewer_yp1c · 2021-07-05
**Detailed analysis on the impact of benchmark standardization across various domains**

**Rating:** 7
**Confidence:** 3

**Strengths:**

Strengths:

- Across both the main paper and appendix, various well known benchmarks were explored and analyzed which help support the core claims made in the paper.
- Many relevant works were cited that cover a wide variety of topics that strengthen the relevance and importance of the paper.
- Section 7 provides useful options for mitigating the lottery effect. In particular, I agree with the importance of multiple train-test splits as a means to calculate score variance and provide a more stable mean score. This is implemented in the OpenML AutoMLBenchmark such that each dataset is evaluated across 10 different fixed train-test splits in a cross-validation scheme which leads to significantly less noisy results when comparing different systems.

**Weaknesses:**

Weaknesses:

- I would have preferred that section 7.2's "Beyond a single dataset." be expanded on as it is central to most of the problems described in the benchmark suites in the paper. I also could not find mention of options for improving comparisons between more than 2 models across a suite of datasets, which can be approached through methods such as average relative rank which ignores absolute metric differences in favor of relative ordering. While having its own set of problems, this method can partially mitigate the problem that different datasets have different performance ranges, while also emphasizing the difference in per-dataset solution quality between two models as the distance in rank between them.

Edit: The authors have added a new section 3.2 to address my concern, however I think that currently its position in the paper is a bit strange and would make more sense if parts were moved to section 7.

**Additional Feedback:**

- It would have been interesting to see some of the suggestions in section 7 applied to some of the existing benchmarks mentioned in the paper (where computationally feasible), such as the different approaches to determining if model A is better than model B across a suite of datasets and measuring the correlation between the approaches.

**Clarity:**

The paper is well written and clear. It is cleanly separated into different sections that discuss relevant aspects of the lottery problem, and many relevant papers are cited. Several minor spelling and grammatical errors are present which may warrant a read-through by the authors to correct, including a missing table reference in the appendix.

**Correctness:**

The paper's claims appear to be correct and genuine, and the solutions proposed seem promising.

**Documentation:**

This paper does not introduce a new benchmark.

**Ethics:**

No, the paper does not introduce a new benchmark.

**Relation To Prior Work:**

Yes. This work is primarily a meta-analysis and thus references many works that contain more focused deep dives on particular aspects that the paper touches on. While previous contributions may have tackled singular topics that the paper discusses, it is my understanding that none discuss the topics as generally and through the lens of as many different fields as this work.

**Summary And Contributions:**

This paper introduces the concept of the benchmark lottery, an idea that what researchers commonly consider the best model at any given point in time is highly dependent on (and fragile to) the particular benchmark datasets considered important by that field. The paper shows that selecting a random subset of these datasets often leads to significant changes in the ranked order of models. Furthermore, the paper investigates the impact of the research community focusing on particular dataset suites leading to overemphasis of techniques that tend to perform well on those suites, and the gradual accumulation of tricks required to be competitive that don't necessarily generalize outside of the benchmark being optimized. The paper also looks at potential improvements to these problems such as standardizing benchmark usage and new benchmark guidelines, and more reliance and consistency in statistical significance testing to avoid model comparisons via point estimates.

---

> ### Author Response · Authors · 2021-07-15
> **Response to reviewer yp1c**
>
> Thank you for your comments and feedback. In the revised version of the paper, we fixed errors in the writing and added a subsection to address some of the comments. Regarding additional empirical analysis where suggestions from section 7 are applied to case studies in the paper, we would like to point out that we did not run any experiments for the case studies we have in this paper and they are based on posthoc analyses of publicly available data from leaderboards or published papers limited to the final score of models on different tasks, which is not necessarily enough for running the suggestions in section 7. However, we agree this is extremely important to see deeper empirical studies and we are actively working on some of these aspects and soon will share more detailed and topic-specific follow-up works.
>
>
> > [...] I also could not find mention of options for improving comparisons between more than 2 models across a suite of datasets, which can be approached through methods such as average relative rank which ignores absolute metric differences in favor of relative ordering [...].
>
> We had a related discussion already under VTAB case study, but as per your suggestion, in the revised version, we add a new subsection “Score and rank aggregation” that discusses several alternatives for aggregation of score/rank over multiples tasks, including geometric mean, rank-aggregation, robust (binned) rank aggregation, and elimination ranking.

---

### Official Review · Reviewer_7Z4S · 2021-07-05
**Raising awareness on the "benchmark lottery" phenomena is important; especially reviewers should be aware of a lot of the points discussed in this paper.**

**Rating:** 6
**Confidence:** 3
**Correctness:** NA
**Clarity:** Yes

**Strengths:**

- Raise awareness of the issues around how we measure progress in machine learning research
- Presents the notion of a 'benchmark lottery'
	- Highlights Task Selection Bias and Community Bias
	- The Community Bias section is especially noteworthy; the gist of this section should be included in the reviewer guideline in the future machine learning venues (hopefully by including more examples from other domains, not just from NLP).
	- I agree with the sub-section "Guidelines for conferences and reviewers"(e.g. "not being SoTA" should be invalid basis for paper rejection.) I hope this becomes a norm not only for NLP conferences but also for machine learning conferences in general.

**Weaknesses:**

- This paper can be improved if it has more case studies, not only on the task selection bias, but also on the statistical testing procedure (i.e. concrete procedures and realistic suggestions to avoid the issues discussed).
- Some sub-fields use crowdsourcing workers to benchmark models (e.g. evaluating how natural machine translations are to humans). Depending on the expertise of evaluators, the performance of models could vary a lot. I think this paper should discuss these issues arising from human evaluation as well.
- Practically, what would be benefited most from this paper is the reviewing process, so I think the paper expands more on the Sec 7.1.


**Additional Feedback:**

I agree that it's important to raise awareness, but it's hard to believe that machine learning researchers were completely unaware of these issues discussed in the paper, and yet the problems have persisted. I feel that the problems stem from the fact that it takes additional/excessive effort and compute resources to do proper evaluation especially for academic environments, rather than not being aware of such issues. For example, performing statistical significance testing to control for every source of variance is prohibitively expensive, especially when it involves re-training a whole neural network model multiple times. A potential solution to these problems is to set up a specialized track in a conference, where the conference assigns papers that claim SoTA for small margins to this specialized track, and perform rigorous evaluations including statistical significance testing on behalf of the authors, in a similar manner that Kaggle competitions are held. There is problem of funding sources, but I think it's worth consideration at this stage where a lot of research papers are incremental.

**Documentation:**

NA

**Ethics:**

No.

**Relation To Prior Work:**

- I think Dacrema et al ("Are We Really Making Much Progress? A Worrying Analysis of Recent Neural Recommendation Approaches", RecSys2019) is an important prior work that's missing.

**Summary And Contributions:**

This paper attempts to raise awareness of the issues arising from benchmarking progress in machine learning research. They demonstrate how performance comparison can be sensitive towards task selection in SuperGLUE benchmark and Visual Task Adaptation Benchmark. They also point out that there is a community bias that determines which subset of tasks and benchmarks are considered to be canonical within the community. At the end, they provide some recommendations to mitigate these issues.

---

> ### Author Response · Authors · 2021-07-15
> **Response to reviewer 7Z4S**
>
> Thank you for your suggestions and comments.
> In the revised version of the paper, we have added the missing references pointed out with the reviewer both in the background section and the subsection discussing the benchmarking in the recommender systems community. We strongly agree that several subtopics touched upon in our paper deserve dedicated attention and extensive (empirical) studies, including the statistical significance testing, while this writeup focuses mostly on the benchmark lottery and expands briefly on the related phenomena and in general, the mission is to call for further discussion. Moreover, we did not run any experiments for the case studies we have in this paper and they are based on posthoc analyses of publicly available data from leaderboards or published papers limited to the final score of models on different tasks, which is not necessarily enough for running the statistical tests. However, we are actively working on some of these aspects and soon will share more detailed and topic-specific follow-up works. Here we reply to some of the questions/comments:
>
> > Some sub-fields use crowdsourcing workers to benchmark models (e.g. evaluating how natural machine translations are to humans). Depending on the expertise of evaluators, the performance of models could vary a lot. I think this paper should discuss these issues arising from human evaluation as well.
>
> We have incorporated the reviewer's suggestion on adding a discussion on the “human evaluation bias” and how cognitive biases and different levels of expertise for the annotators can contribute to the benchmark lottery.
>
> > Practically, what would be benefited most from this paper is the reviewing process, so I think the paper expands more on the Sec 7.1.
>
> We have added a benchmarking checklist (as per one of the reviewers’ suggestion) to the Appendix, that lists some concrete checks for improving the review process.
>
> > Additional Feedback [...]
>
> The suggestion for “setting up specialized tracks at conferences that offer help in terms of expertise, infrastructure, and computational resources for extensive evaluation to the papers submitted to that conference” is a great way for facilitating the rigorous testing of claimed improvements. We have expanded a bit on this and added this suggestion to the revised paper. Thank you for sharing the idea.

---

### Author Response · Authors · 2021-07-15
**Thanks to all the reviewers**

We would like to thank the reviewers for the time they spent reading through our submission and for the great comments, feedback, and suggestions. We have revised the submission by: updating several parts to address some suggestions and comments, fixing some errors in the writing, updating the checklist with more explanation, solving issues with rendering the plots, and adding some missing/new references. We have also added some parts to summarize the ideas and expansions that were proposed by reviewers. We will reply to reviewers' comments with detailed explanations.

---

### Public Comment · ~Yosuke_Shinya1 · 2021-07-27
**About rigging the lottery**

Hi authors!
Sec. 6 and Sec. 7.1 relate to our work "USB: Universal-Scale Object Detection Benchmark" https://arxiv.org/abs/2103.14027 (Although the arXiv submission is recent, its preliminary work was presented at a CVPR 2020 workshop). Our paper discusses unfairness mainly by resource-intensive settings: long training epochs and hyperparameter optimization (please see Sec. 1 and Sec. 3.3). They are "expensive lotteries", which are more problematic than inexpensive ones because only some well-funded researchers can buy the lotteries. In my opinion, the discussion on long training epochs could be added around that on hyper-parameter tuning budget.

Furthermore, before giving concrete examples (Sec. 6.1), it would be better to add an abstract and macro-perspective summary of lottery factors (e.g., task lottery, dataset lottery, split lottery, metric lottery, stochastic training lottery, mini-batch lottery, architecture lottery, hyperparameter lottery, parameter lottery, software lottery, hardware lottery). I think a table like below improves the value as a survey paper.

|Lottery factors|Problem description|Reference|
|:-:|:-:|:-:|
|parameter lottery|| Lottery Ticket Hypothesis https://arxiv.org/abs/1803.03635 |

---

### Decision · Program_Chairs · 2021-07-26

**Decision:**

Reject

**Comment:**

The reviewers generally liked this paper, but in the post-rebuttal discussion there was unanimous consensus that the paper does not feel ready for publication in this selective venue. In particular, the many typos (still in the updated version), references to removed text, opinions instead of facts, missing code, etc. There was also confusion amongst the reviewers whether the authors actually ran experiments themselves or only reported on others' experiments. Indeed, the reviewers assumed the former until a rebuttal comment, and even remained confused at that point due to the text in the paper (lines 149-150: "We have evaluated 32 different models against all the 19 VTAB tasks.")
Other feedback from the post-rebuttal discussion:
- It is disappointing that even in the rebuttal the authors only promise that the code will be available rather than taking the chance to make it available — this does not appear in line with the practices of reproducibility this track aims to improve. (This is important even if this is only code for analyzing results, and that code should be even much easier to share than code for running experiments.)
- Part of section 3.2 does seem like it could belong to Section 7.
- The new set of best practices in benchmarking that the authors now propose in Appendix F could be improved a lot.

At the end, even the reviewer who gave a 7 said that he/she now feels inclined to reject.

I encourage the authors to polish the paper and resubmit.